# Wildfire legacies on pyrogenic carbon stocks in Amazonian peatlands
Yuwan Wang [1,2] ✉, Angela Gallego-Sala [1], Michael I. Bird [3], Patrick Moss[2,4], Hamish A. McGowan [2], Juan C. Benavides [5], Eurídice N. Honorio Coronado[6] & Ted R. Feldpausch [1] ✉

Amazonian peatlands are carbon-rich ecosystems that act as long-term carbon sinks but have faced increasing fire risks in recent decades. As a legacy of past fires, the contribution of pyrogenic carbon (PyC) to carbon cycling in these peatlands remains poorly understood. Here, we assess PyC accumulation variability using six cores spanning peatlands in northwestern Amazonia using hydrogen pyrolysis. We also estimate the PyC stock for the entire Amazonian peatlands by combining our field dataset with published sources. The PyC to total organic carbon ratio averaged 1.2% across our sites and increased with peat age. We estimate a total peatland PyC stock of 0.73 ± 0.61 Pg for the Amazon Basin, representing 1.6% of their TOC stock. Due to the slower turnover of PyC in peatland ecosystems, our findings indicate the importance of PyC generated by past fires and highlight the potential long-term carbon sequestration role of PyC in the future carbon cycle.

The Amazon Basin contains ~150–200 Pg C in biomass and soil[1], representing a carbon pool of global importance[2]. In recent years, increased drought-related wildfires have led to significant loss of carbon from human-modified landscapes, and this carbon loss also extends to intact forests during extremely dry years[3–5]. Peatlands are one of the most carbon-dense ecosystems within the basin and store substantial carbon in the underlying layers, which have developed in waterlogged, anaerobic conditions over decades to millennia[6,7]. Undisturbed Amazonian peatlands are not necessarily fire-prone, as they maintain almost permanent waterlogged conditions in their soils and exist mostly in areas of sustained rain throughout the year. However, fire risk can change once peatlands are impacted by land use change or droughts[8]. Though the current human activities on Amazonian peatlands have limited impact, ongoing large-scale land conversion to meet commercial and subsistence needs[9] combined with intensified climate anomalies[10,11], represent an increasing fire risk to this long-term carbon sink. In other parts of the tropics, large-scale drainage and land use changes for oil palm plantations and agriculture in Southeast Asia have made this peatland-rich region susceptible to fires[12]. The El Niño Southern Oscillation phenomenon induced a prolonged dry season in Indonesia in 1997, drying the already disturbed peatlands to such an extent that they burned, releasing between 0.81 and 2.57 Pg C[13].

During wildfires, pyrogenic carbon (PyC), which includes a wide range of sizes and compounds of pyrogenic origin, such as slightly charred biomass, black carbon and soot, is simultaneously produced from the incomplete combustion of biomass[14,15]. Owing to its persistence in the environment, PyC is divided into labile, semi-labile, and stable fractions, representing the degradation time of years, decades, and centuries to millennia, respectively[14]. While most PyC remains initially on the site of burning, it can later be partly relocated or distributed through terrestrial and marine ecosystems by erosion and/or leaching[16,17]. Additionally, a portion of aerosol-sized PyC can be dispersed in smoke plumes over long distances[14,18]. The temperatures of fires in tropical forests and savannas can exceed 500 °C[19,20]; under these conditions, 80% of the PyC produced is likely to exist in a relatively stable form with highly condensed aromatic ring structures[14]. Because of the slow decompositional anoxic environment in intact peatlands, both stable and more labile PyC generated from peatland and/or wider landscape burning can be preserved for longer compared to soils that are exposed to oxic conditions. Thus, PyC stored in peatlands may represent an important slow-cycling fraction as a legacy of past fires.

PyC has been largely ignored in global soil carbon estimates over the years; however, a growing number of studies have focused on this carbon pool, and these have emphasised its global importance. The most recent PyC estimates suggest that the global PyC stock lies within the range of 80–450 Pg C[21], which equals ~5–26% of the total soil carbon stocks globally (1700 Pg[22]). This range is comparable to previously reported values[14,23,24]. Yet, until

[1]Department of Geography, University of Exeter, Exeter, UK. [2]School of the Environment, University of Queensland, Brisbane, Australia. [3]College of Science and Engineering and ARC Centre of Excellence for Indigenous and Environmental Histories and Futures, James Cook University, Cairns, Australia. [4]School of Earth & Atmospheric Sciences, Queensland University of Technology, Brisbane, Australia. [5]Department of Ecology and Territory, Pontificia Universidad Javeriana, Bogotá, Colombia. [6]Royal Botanic Gardens, Kew, Richmond, London, UK. ✉e-mail: yw637@exeter.ac.uk; t.r.feldpausch@exeter.ac.uk

recently, large uncertainties and limitations have remained when estimating global PyC stocks. This mainly stems from a lack of agreement between various methods used to estimate PyC stocks and a limited exploration of controls on the spatial and vertical distribution in PyC, especially in tropical regions where the data is generally sparser than in temperate or higher latitude regions[23]. This is the case even though biomass burning in the tropics was the source of ~91% of fire-induced PyC between 1997 and 2016[21].

Fire regimes vary across the tropics, and this impacts PyC production. For tropical regions, a PyC to total organic carbon ratio (PyC/TOC) of 15% is assumed for the most flammable moist and dry ecosystems, while in the least flammable, tropical wet ecosystems, values of 1% PyC/TOC have been suggested[14]. The latter amount was considered the case for the majority of the Amazon Basin, especially for the wettest northwestern portion[11,25]. However, this percentage is likely to be an underestimation, especially for deep soil layers, where TOC is low[26–28]. A more recent study[26] reported an average PyC storage of 11% of TOC for the 0–200 cm soil depths in the whole Amazon Basin, based on sampling from permanent plots in *terra firme* natural forest (non-*terra preta*) with no history of recent fire. This study was the first basin-wide PyC estimate and revealed an increasing PyC/TOC trend down to 2 m as TOC declined faster with depth than PyC. This emphasises the importance of PyC in TOC stored in deeper layers.

There are very few studies estimating peatland PyC. The only study of PyC stored in peatlands is from high-latitude northern peatlands and reported a total PyC stock of ~62 Pg based on PyC-age relationships[29]. The age of the peat provides information on long-term apparent rates of carbon accumulation. This facilitates the temporal comparison of carbon dynamics for large datasets on the regional scale[30] and possibly allows linking broader climatic controls at the same temporal scale[31].

Peatlands in Amazonia were first documented in the 1950s[32], and scientific studies on peat properties have only recently been conducted[6,33–37]. Insights into Amazonian peatlands stem mainly from studies of the Peruvian peatlands, with the Pastaza-Marañón Foreland Basin (PMFB) being considered the largest peatland complex identified so far[7,35,37]. Many areas remain largely unexplored, mostly due to accessibility constraints, such as the Rio Negro Basin in Brazil, where extensive peatlands could potentially exist[38,39]. Although some progress has been made in understanding carbon cycling in lowland Amazonian peatlands, such as those in Peru, based on intensive fieldwork[36], our understanding of specific processes, such as PyC accumulation and dynamics, is very limited.

To our knowledge, there have been no studies of PyC storage in Amazonian peatlands. An accurate estimation of PyC in the Amazon Basin that accounts for peatlands is required to broaden our understanding of the role of PyC in influencing future regional and global carbon dynamics. In this study, we use hydrogen pyrolysis (HyPy) to quantify PyC in six peat cores from northwestern Amazonia and further upscale the results to estimate the basin-wide peatland PyC stocks based on available peat properties (peat depth, dry bulk density, carbon content and peat basal age). This study addresses the following questions: (a) Is there an age-dependent vertical distribution of PyC in the studied peatlands, and if so, what is the pattern? (b) What is the PyC stock in basin-wide peatlands, representing the total PyC accumulated during peat formation in the Amazon Basin? (c) Does PyC storage in peat soils differ from forest soils in the Amazon Basin, considering the area of each soil type?

## Results

### TOC and PyC variability with age

TOC and PyC/TOC varied significantly between sites (Kruskal–Wallis test, $p < 0.001$). TOC values ranged from 16.1 to 62.2% with an average of 38.3% across all sites (Table 1), while PyC/TOC ranged from 0 to 17.1% with a mean value of 1.2%. INI, the northernmost site (Fig. 1) located in flooded forests, had the highest average PyC/TOC values, while CAN (*Mauritia*-dominated palm swamp), as the closest site to the centre of the Amazon forest, exhibited the lowest levels, approximately 300 times lower than INI. Zero values are considered an indication of no fire and are observed for more

than half the time for the entire fire history at two *Mauritia*-dominated palm swamps: CAN (spanning ~600 years) and QUI (spanning ~2600 years), suggesting that fire has been infrequent at these sites and/or the surrounding areas. Significant spatial autocorrelation was found in PyC/TOC (Supplementary Fig. 1a), while a cluster of low values appeared to be located at southern sites (QUI, JEN, and CAN) (Supplementary Fig. 2). No spatial autocorrelation in TOC was found (Supplementary Fig. 1b).

The overall trend between TOC and age was non-significant, suggesting that TOC values were relatively stable across the vertical profile. Only QUI and CAN showed significantly negative correlations among these sites. PyC/TOC increased with peat age in the simple linear model across all sites, and the increasing trend remains in the linear mixed-effect model, when accounting for the influence of sites (Table 2; Fig. 2). The trend of PyC/TOC increasing with age was observed in most sites, suggesting that PyC is relatively stable than non-PyC fractions in TOC and can be effectively preserved in older, deeper peat layers. No spatial autocorrelation in the trend of PyC/TOC was observed (Supplementary Fig. 1c).

### TOC and pyrogenic carbon stocks

We estimated a total TOC stock of $46.7 \pm 31.4$ Pg in Amazonian peatlands, of which $10.2 \pm 6.4$ Pg was estimated in northwestern Amazonia. These estimates were based on the peatland area within the Amazon Basin ($462,833$ km$^2$ [39]), peat depth ($183.9 \pm 116.1$ cm) obtained from 1095 observations at 127 sites, and peat bulk density ($0.13 \pm 0.07$ g cm$^{-3}$) and carbon content ($40.8 \pm 9.5\%$) calculated from 344 and 263 observations from 53 and 43 sites, respectively (Fig. 3; Supplementary Table 1). Considering the peatland extent in these areas, we obtained a TOC stock of $1008 \pm 678$ Mg C ha$^{-1}$ for peatlands in the whole Amazon Basin and $983 \pm 613$ Mg C ha$^{-1}$ for peatlands in northwestern Amazonia alone. Because our datasets are derived mainly from the observations from northwestern Amazonia with extensive data compiled within the area of the PMFB (Fig. 3), the carbon stock obtained from the compilation is within the range of three most common peatland ecosystems discovered in PMFB: pole forest, palm swamp and open peatland (the average value of $892 \pm 535$ Mg C ha$^{-1}$ for PMFB and $1391 \pm 710$ Mg C ha$^{-1}$ for pole forest, the most carbon-dense ecosystem among three ecosystem types)[35].

Using a median peat initiation age of 4450 cal yr BP for the whole basin and 2537 cal yr BP for northwestern Amazonia (Supplementary Table 2), we calculated a PyC stock of $0.73 \pm 0.61$ Pg for all Amazonian peatlands and $0.13 \pm 0.10$ Pg for peatlands located in northwestern Amazonia, which is equivalent to 1.6% and 1.2% of the corresponding peatland TOC stocks, respectively.

## Discussion

While there have been studies on measuring PyC/TOC in soils, more than half of these observations are limited to the surface layer, as evidenced by a compilation study[23]. Recent efforts to investigate the deep storage of PyC[26,40] demonstrate its notable contribution to the carbon pool in forest soils. In peatland ecosystems, large amounts of carbon accumulate throughout all layers, compared to forest soils. The presence of carbon-rich deep layers is more common in peatlands, as seen in Peru, where thick peat can reach up to a depth of 7 m and the average peat thickness is ~2 m[37]. PyC, as the result of peatland burning or wider landscape burning, can be well preserved with peat materials under anoxic conditions. However, there is very little knowledge about this deep store of PyC in peatlands. Our analysis of PyC storage in Amazonian peatlands reveals a much higher PyC stock per unit area than in forest soils in the basin. The calculations of basin-wide PyC peatland stocks are based on northwestern Amazonia, where fires have been rare due to very wet conditions and limited human intervention; thus, this storage estimate may be an underestimation. Our findings highlight the critical role of PyC sequestration in carbon-rich peatland ecosystems and suggest the necessity of incorporating PyC in peatlands into regional and global carbon dynamic models. This is particularly the case because the residence time of PyC in peatlands (and mineral soils) is very different from non-PyC fractions in TOC, and this needs to be explicitly taken into account in these models.

**Table 1 | Average and standard deviation (SD) of TOC and PyC/TOC for each site**

| Site | TOC (%) | | | PyC/TOC (%) | | |
|---|---|---|---|---|---|---|
| | **Mean** | **SD** | **Slope with age** | **Mean** | **SD** | **Slope with age** |
| INI | 31.4 | 6.4 | −0.0008 | 9.07 | 5.39 | 0.0009 |
| PLL | 24.6 | 4.9 | 0.0013 | 1.68 | 0.39 | −0.0001 |
| CAQ | 31.0 | 11.4 | 0.0017 | 0.88 | 0.66 | 0.0004 |
| QUI | 56.6 | 4.4 | −0.0031* | 0.13 | 0.22 | 0.0002* |
| JEN | 24.1 | 5.8 | 0.0127 | 0.50 | 0.17 | 0.0028* |
| CAN | 41.2 | 10.6 | −0.0421* | 0.03 | 0.05 | 0.0000 |
| Overall | 38.3 | 14.4 | −0.0013 | 1.19 | 2.70 | 0.0012* |

Slope with age obtained from simple linear regression of TOC and PyC/TOC for individual sites (* indicates $p < 0.05$).

## Pyrogenic carbon stock in Amazonian peatlands

We provide a first PyC estimate in Amazonian peatlands, equivalent to 1.6% of the peatland TOC stock, while the estimated value for northwestern Amazonia is 1.2%. These values fall within the global range of 0-60% in soils, below the global average of 13.7%[23], and are consistent with the previously estimated values of 1% for tropical wet regions with infrequent fire[14]. Previous studies of PyC in forest soils in the Amazon Basin showed a similar order of magnitude to our study, ranging from 1.3 to 7.3%[26–28,40]. Compared to the global average of 13.7%[23], these low PyC values in the Amazon Basin reveal the generally low incidence of fire in the Amazon Basin in a global context.

With a high TOC stock in Amazonian peatlands, our best estimate of PyC stock in these peatlands (16 Mg ha$^{-1}$) is four times higher than forest soils in the basin (Table 3). Our peatland PyC stocks are much smaller than those reported for high-latitude northern peatlands[29], which could be linked to either the higher flammability of peat in high-latitude peatlands or differences in fire regimes for these regions[7,41]. The generally high-water content of undisturbed peatlands prevents wildfires from occurring most of the time. However, whenever peat becomes dry, it can burn for long periods as smouldering ground fires can occur at depth[42], with dry peat layers acting as additional fuel. For example, during the 1997 peatland fires in Indonesia, peat combustion to a depth of 51 cm was reported[13]. During such fires, large amounts of PyC can be expected to be produced from its precursor biomass and buried in the peat[43]. Although newly produced PyC globally accounted for ~12% of carbon emissions between 1997 and 2016, which may benefit long-term carbon sequestration, the combustion of deep organic soils, such as in degraded peatland ecosystems, was not included in this assessment[21]. Given the considerable carbon stocks stored in peatlands and the large emissions that could be released in the short term when drained, mitigating related emissions from peatlands could be achieved by preserving these intact peatlands, combined with effective fire management.

## Pyrogenic carbon and age: vertical variability

The general increase in PyC/TOC values with peat age in our study indicates that PyC is playing a progressively more important role in the deeper, older layers compared to other forms of TOC, although the trend is not significant, possibly due to the small number of PyC/TOC observations. A similar increasing trend was observed in intact high-latitude northern peatlands[29], as well as in soil ecosystems to a depth of 1–2 m at similar latitudes, in Amazonia *terra firme* forests[26] and in intact forests and forest fragments located in the northern Brazilian Amazon[27,40]. A similar result has also been observed in soils at higher latitudes, such as on the Chinese loess plateau[44]. However, no significant difference between depths was detected in a comprehensive global compilation of PyC/TOC that gathered 569 individual PyC/TOC values from 55 different studies across different ecosystems[23]. That study used only topsoil (top 10 cm) and subsoil (the depth below 10 cm) as the categorical factors rather than considering a continuous depth profile with only 2 sites identified as peat. Our current knowledge of this possible trend, therefore, remains uncertain due to the limited number of studies on peatland ecosystems.

On a regional level, aligning with the increase in PyC/TOC ratios with peat age in most of the studied sites, higher charcoal values between 2500 and 2000 cal yr BP in lake sediment compared to more recent ages in northwestern Amazonia were observed[45]. Radiocarbon dating of charcoal in *terra firme* forests in northern and southern Peru showed a total of four to six unique fire events over the past 2000–4000 years, with no fires occurring within the past ~700–850 years[46]. Charcoal abundance in the American tropics showed a general increase from 3 to 2 ka, followed by a decline into the industrial era[47], suggesting more frequent fires in the millennia before the industrial age. This pattern could be associated with the wetter climate in the Amazon Basin during the late-Holocene, coupled with the southward movement of the Intertropical Convergence Zone, boosting the intensity of the South American summer monsoon and bringing more rainfall into the basin[48,49]. For example, a continuous rise in water levels was documented in Lake Titicaca, a high-altitude lake in the central Andes, over the last 4000 years[50].

After PyC is produced, its vertical distribution will depend on preservation and stabilisation processes in sediment and soils[51,52]. Oxygen availability is a key factor regulating the degradation of all carbon molecules, including PyC, which will eventually result in different preservation potential in peatlands and mineral soils. Due to the waterlogged conditions that are pervasive in peatlands, oxygen concentration, especially at depth, tends to be low, while the surface peat, with greater input of fresh organic matter, can be exposed to more aerobic conditions. This favours higher microbial activity in the upper layers of tropical peatlands[53,54]. Additionally, microbes preferentially degrade other forms of carbon that can be more easily used. Therefore, PyC will be selectively preserved over time and undergo relative accumulation in peats[52]. The downward movement of PyC, including the translocation of the particulate PyC form and dissolved PyC form as a result of leaching, could also enrich PyC with depth. This downward transport has been observed in mineral soils[17,51] and in drained peatlands where high movement rates of 0.6–1.2 cm yr$^{-1}$ have been reported[55]. These possible mechanisms may together contribute to the higher PyC in the study at greater depth and over time. The lack of a spatial pattern in PyC/TOC over time in our study sites may indicate that each site is differently influenced by local conditions (e.g., hydrology, topography, remote PyC sources, and peat accumulation rates) and these local factors are more important than regional phenomena in driving PyC spatial patterns. This issue needs to be investigated in future research to gain a clearer understanding of the main drivers of PyC accumulation.

## Spatial variability of pyrogenic carbon

PyC directly reflects the incidence and intensity of fire in an ecosystem, and the occurrence of fires in tropical peatlands is normally related to alterations in hydrological conditions[8]. In recent years, increased fire largely coincides with a combination of climate and human-induced drying[56], while in the past, with limited human impact, it is typically linked to drought events[57]. Peatlands located in northwestern Amazonia have experienced less severe drought stress during the El Niño Southern Oscillation-induced basin-wide drought events that occurred in 1982–1983, 1997–1998, 2015–2016[58], and 2023–2024[59]. This lower fire activity may have been the case for much of the Holocene compared to other regions in the Amazon Basin, as demonstrated by charcoal records from lake sediments[45]. The median fire return interval (~450 years) from *terra firme* Amazonian forest sites in seasonally wetter northern and seasonally drier southern sites in Peru[46] also suggests less fire over past millennia for the forests in western Amazonia. Hence, the PyC/TOC values for the entire basin here generated mainly from the northwestern region are only likely to underestimate PyC/TOC in the wider Amazonian peatlands when they share similar peat inception ages.

The different PyC values across our study sites are likely attributable to varying fire histories at the locations or in their surrounding environment. This is reflected in the higher PyC/TOC values at sites closer to the natural

**Fig. 1 | Study sites in the Amazon Basin.**
**a** Locations of study sites (white triangles) on the map of averaged Maximum Cumulative Water Deficit (MCWD) from 1981 to 2020[83] and **b** on the land cover map[84]. White lines indicate regional boundaries[85,86]. NW Northwestern Amazonia, SW Southwestern Amazonia, GS Guiana Shield, EC Eastern & Central Amazonia, BS Brazilian Shield. Green lines refer to the Pastaza-Marañón Foreland Basin (PMFB)[87].

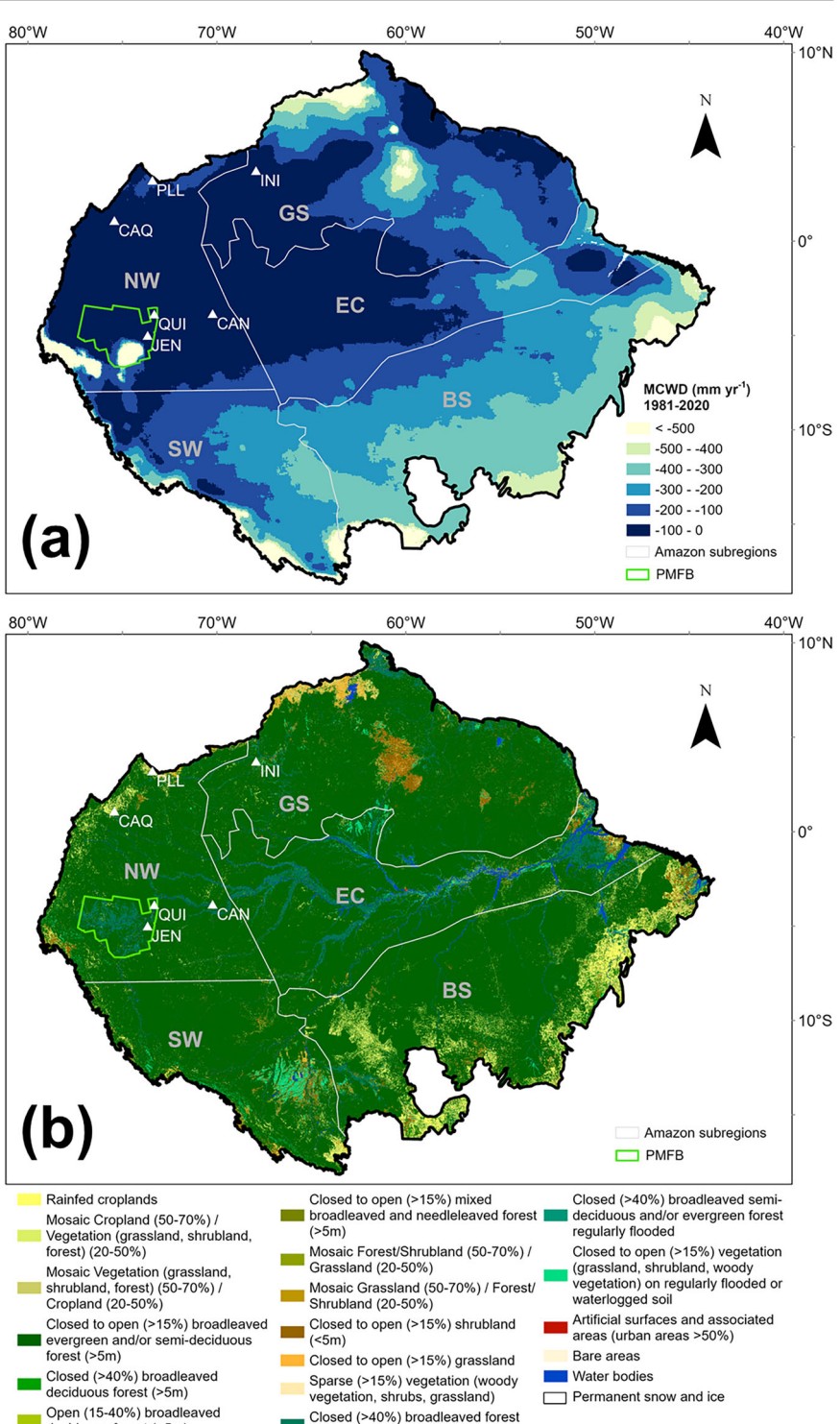

forest limits in INI, PLL and CAQ, compared to those located in the central part of the forest cover (QUI, JEN, and CAN) (Fig. 1). The forest edge is surrounded by more fire-prone ecosystems, such as savannas. Aerosol deposition from smoke from surrounding areas or locations that are further afield, and for example as far away as Africa, could bring additional PyC into the Amazon Basin, although this process is restricted by distance and seasonality[60]. Yearly fire burn fraction (per grid cell) between 1997 and 2016, from Global Fire Emissions Database with small fires[61] shows more frequent burning in our northern sites and surrounding areas that are closer to the edge of the basin than QUI, JEN, and CAN, where the lower clusters of PyC/

TOC were observed (Supplementary Fig. 2). This suggests peatlands that are surrounded by drier environments at the edge of the Amazon Basin (e.g., areas adjacent to the Cerrado or Llanos) potentially receive higher PyC inputs from nearby fire-prone environments, which contribute to higher PyC deep storage at those sites.

*Mauritia*-dominated palm swamps of three southern sites (QUI, JEN, and CAN), presented significantly lower PyC values compared to all the other ecosystem types in this study, including mixed palm swamps, palm swamps and flooded forest (Supplementary Fig. 3). It is worth noting that PyC/TOC values in the INI site are much higher than in the other sites,

**Table 2 | Regression models of PyC-age relationships with and without site effects**

| Models for PyC/TOC | | Variables | ΔAIC* | Intercept | Slope | $R^2$ |
|---|---|---|---|---|---|---|
| *Models with all ages (N = 67)* | | | | | | |
| Model 1 | linear mixed effects model with random slope | age, site (random factor) | 0.0 | 0.80 | 0.34 | 0.78 |
| Model 2 | linear mixed effects model with random intercept | age, site (random factor) | 9.9 | 0.57 | 0.81* | 0.83 |
| Model 3 | simple polynomial model | age | 16.8 | 0.11 | 0.47* (1) 0.08* (2) | 0.77 |
| Model 4 | simple linear model | age | 26.9 | −0.36 | 1.16* | 0.72 |
| *Models up to 3500 cal yr BP (N = 64)* | | | | | | |
| Model 1 | linear mixed effects model with random slope | age, site (random factor) | 0.0 | 0.85* | 0.43 | 0.96 |
| Model 2 | linear mixed effects model with random intercept | age, site (random factor) | 51.1 | 0.97 | 0.43* | 0.91 |
| Model 3 | simple polynomial model | age | 82.9 | 0.52* | −0.97* (1) 0.58* (2) | 0.42 |
| Model 4 | simple linear model | age | 97.5 | 0.08 | 0.64* | 0.26 |

*ΔAIC indicates the best model in ascending order.
The age applied to the equation is ka (millennium scale). (*indicates $p < 0.05$).

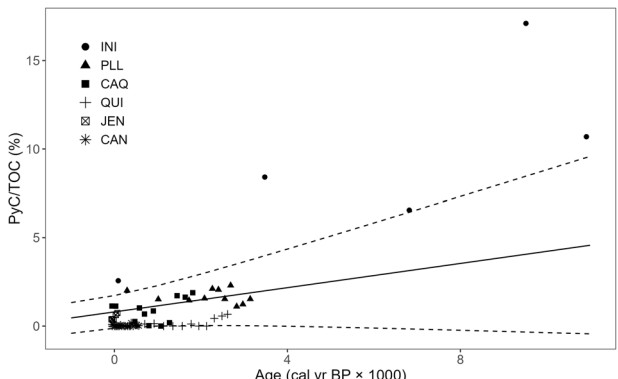

**Fig. 2 | The relationship between the percentage of pyrogenic carbon in total organic carbon and peat age in Amazonian peatlands ($N = 67$).** The shape of data points indicates different peat cores. The solid line is the best model derived from a linear mixed-effects model with random slope (Table 2). The percentage of PyC/TOC increased by 0.34 (95% CI: −0.08 to 0.76) per ka with the intercept 0.80 (95% CI: −0.12 to 1.73). Dashed lines indicate a 95% confidence interval.

methodological difference[66], our HyPy-derived PyC/TOC still indicates lower values in peatlands compared to reported values in Amazonian forest soils. Furthermore, the lack of accurate and detailed peat maps across the basin could add extra uncertainties, although some regions where field campaigns have been more extensive have more reliable maps[36,37]. The recently published field data-driven peatland map for the basin[38] would provide a lower estimate of 0.40 Pg of PyC (based on a different boundary of the Amazon Basin from our study). Uncertainties remain regarding the coupling between PyC/TOC and peat inception ages. Addressing these data gaps through future research is needed to improve the accuracy of PyC stock estimates in Amazonian peatlands.

The conservation of intact peatlands has been key to reducing fires and controlling PyC dynamics within the basin. Extensive construction projects are planned for peat-rich areas of the Amazon Basin[9], where PyC would become more prevalent due to the high flammability resulting from lowered water tables in the potentially affected areas. Our findings that PyC/TOC is relatively low in intact Amazonian peatlands, both regionally and globally, suggest that these intact ecosystems are naturally resistant to wildfire. Area-based PyC stocks in Amazonian peatland ecosystems are higher than forest soils in the basin and lower than reported for high-latitude northern peatlands. This highlights Amazonian peatlands' crucial role in the local PyC budget for short periods and over millennia; however, their distinctive character differs from high-latitude northern peatlands. The reconstructed historical fire patterns provide insights into trajectories of the fire frequency and carbon cycling in these water-logged environments under changing land use and increased drought severity in a drier future Amazon[10,11].

which may partly be explained by this site experiencing more variable water levels in flooded forest than in palm swamps[62]. Therefore, there is greater potential for being affected by lower water levels during droughts or prolonged dry periods, which would increase the likelihood of local fires occurring.

## Uncertainties and future research
Our dataset provides an important step towards understanding the role of PyC in carbon storage in Amazonian peatland ecosystems; some uncertainties and limitations could be addressed with future work (see Supplementary Discussion for details). To estimate peatland PyC stocks, we assumed that TOC is equally distributed across the peat vertical profile and that the median value from our compilation of peat basal ages represents the general age of peat inception across the basin. These two assumptions were based on TOC values ($N = 3741$) from high-latitude northern peatland[30] and a global compilation of peat basal ages ($N = 1097$)[63], providing a conservative estimate of PyC stock. Comparing PyC results generated by different quantification methods is challenging due to non-systematic offsets between the current approaches[64,65]. However, considering this

## Methods
### Sample collection
There is pronounced hydroclimatic variability within the Amazon Basin, with precipitation during the three driest months of the year following a geographically decreasing trend from the northwestern basin towards its southeastern portion[67]. Our study region is located in northwestern Amazonia, which is characterised by a tropical rainforest climate[68] that experiences the least climatological water deficit throughout the year of anywhere in the entire basin (Fig. 1a).

One core was collected from each of the six lowland peatland sites in Colombia (Inírida-INI, Puerto Lleras-PLL, Caquetá-CAQ and Cananguchal-CAN) and northern Peru (Quistococha-QUI and Jenaro Herrera-JEN) between 2018 and 2022 using a Russian D-section corer to the base of the

**Fig. 3 | Spatial distribution of peatland properties across the Amazon Basin.** Maps show **a** peat depth (cm), **b** dry bulk density (g cm$^{-3}$), **c** carbon content (%) and **d** peat basal age (cal yr BP) in the Amazon Basin (outlined by thick black border) with the peat map[39]. The inset in each panel refers to the Pastaza-Marañón Foreland Basin (PMFB)[87], the most extensive peatland area in the basin. Full datasets and references for this figure are listed in Supplementary Tables 1 and 2.

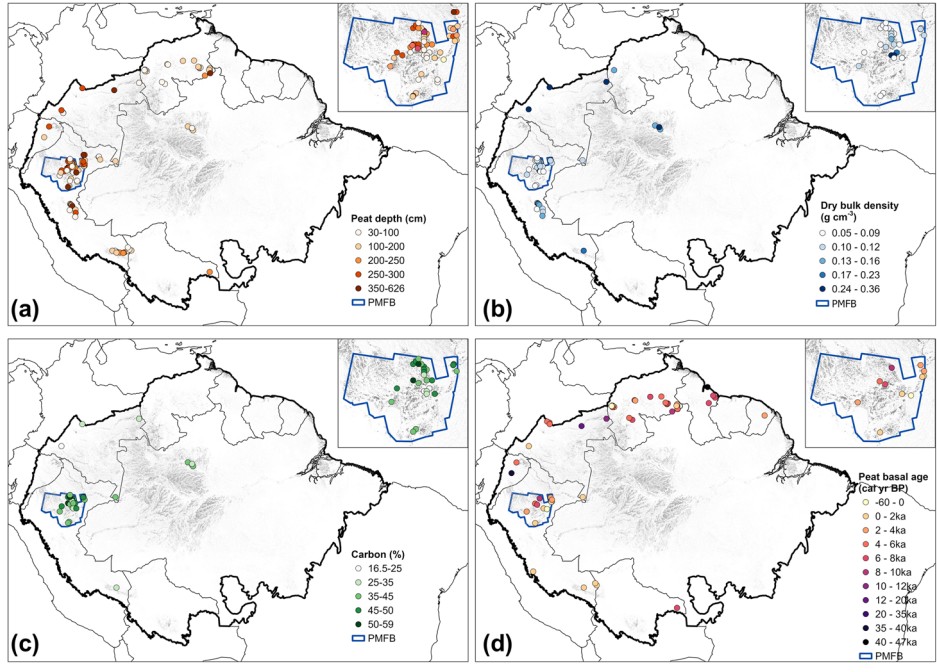

peat or the maximum depth that could be reached in the field (Supplementary Table 3). All sites are permanently water-logged, and classified as *Mauritia*-dominated palm swamps (CAN, QUI and JEN), mixed palm swamps (CAQ), palm swamps surrounded by agricultural land (PLL) and tree-dominated swamps within flooded forests (INI) based on their modern vegetation. These peatlands have maintained their integrity and have not experienced land use changes or alterations of their hydrological conditions, although selective harvesting of the most dominant species (*Mauritia flexuosa*) could have occurred in the past at the Peru sites[69]. There have been no reports of recent fires at any of our sites.

### Chronology

Twenty sub-samples at different depths covering the length of each of the six cores were prepared for AMS $^{14}$C analysis (Supplementary Table 4). For INI, PLL, CAQ, and CAN, bulk samples were sieved through a 63 µm mesh, and the <63 µm fraction underwent hydrochloric acid pre-treatment for analysis at the $^{14}$CHRONO Centre in Queen's University Belfast. For QUI and JEN, macrofossil samples were hand-picked and submitted to the National Environmental Isotope Facility (NEIF) Radiocarbon Laboratory and analysed at the Scottish Universities Environmental Research Centre (SUERC) AMS Laboratory with acid-alkali-acid pre-treatment. All $^{14}$C dates were reported as uncalibrated year before present (yr BP, present = 1950 CE) and then calibrated using CALIB 8.2[70] with a 50:50 mix of the IntCal20 and SHCal20 mixed calibration curve due to the influence of the Intertropical Convergence Zone[71].

$^{210}$Pb measurements were performed only for QUI and JEN at contiguous 1 cm intervals for the topmost peat layers (Supplementary Table 5). Samples were spiked with $^{209}$Po, digested with HNO$_3$–H$_2$O$_2$–HCl and electroplated onto silver planchets. $^{210}$Pb activity was determined using alpha spectrometry, measuring the decay of $^{210}$Po, the daughter product of $^{210}$Pb, at the Geography Radiometry Laboratory, University of Exeter. Bayesian chronological models were used to construct age-depth relationships from the calibrated $^{14}$C dates using the *rbacon* package[72]. For QUI and JEN, the *rplum* package was used to combine the different dating methods, $^{210}$Pb and $^{14}$C dates[73]. These two packages were built by the same authors, and thus, there remains a degree of overlap in the methodology. For PLL and INI, due to age reversal at deep depths, only dates for the top 300 cm and 42.5 cm were used for age-depth models, respectively (Supplementary Fig. 4).

### Pyrogenic carbon quantification

For PyC analysis, ~200 mg bulk samples from 5 to 20 cm intervals, depending on the total length of cores, with a total of 67 samples, were weighed and treated overnight with 2 M HCl to remove any potential inorganic carbon. Samples were then water-washed, dried, and hand-ground into a fine powder. The PyC analysis was carried out in the Advanced Analytical Centre at James Cook University following the standard procedure for HyPy[74]. Briefly, the powdered sample was loaded with ~10% Mo catalyst and heated to 250 °C at 300 °C min$^{-1}$, then 8 °C min$^{-1}$ to 550 °C and kept for 2 min under a 150 bar of hydrogen with 5 L min$^{-1}$ sweep gas flow in the reactor. Stable PyC with aromatic rings greater than seven is isolated for analysis using this technique[66]. The carbon content before and after HyPy was measured using an Elementar PrecisION IRMS at the Stable Isotope Geochemistry laboratory, The University of Queensland, and was then used to calculate PyC abundance. Final PyC values were corrected to eliminate the effect of in situ PyC production in the HyPy reactor[74] and reported as PyC/TOC.

### PyC-age model selection

To assess how PyC/TOC varies with the age of the peat and to apply the generated relationships to a wider geographic range, simple linear and polynomial regressions were applied via the *stats* package[75]. Linear mixed-effects models for each site over age via the *lme4* package[76] were also used.

The best model was determined by evaluating the Akaike Information Criterion (AIC) index (the smaller, the better) with ΔAIC = 0 using the *bbmle* package[77]. We compared four models for two different time periods by: (1) keeping all observations (*N* = 67); and (2) only considering PyC/TOC up to an age of ca. 3500 cal yr BP, as there was only INI data older than this date (*N* = 64) (Table 2). Notably, for both time periods, the slope in most models, except for model 1 (the best-performing model), was significant, indicating PyC/TOC increased significantly with increasing age. However, the disappearance of significance in the best-performing model (model 1, which includes site as a random factor) suggests that the significance could be mainly driven by site differences.

To test the robustness of our models, we used a rank-based linear model implemented via the *Rfit* package[78] with non-parametric analysis, and the slope shows a similar but slightly larger magnitude (0.46% PyC/TOC ka$^{-1}$) to the best-performing models under both conditions. To capture the relationship between PyC/TOC and age for the longer period, we chose to continue further analyses using the best-performing model with all

**Table 3 | Estimates of PyC stock and PyC/TOC in Amazonian and high-latitude northern peatlands and Amazonian forest soils**

| Ecosystem type | Study region | PyC stock based on area (Mg ha$^{-1}$) | PyC/TOC (%) | PyC method* |
|---|---|---|---|---|
| Peatland | Amazon Basin (this study) | 16 | 1.6 | HyPy |
| | Northwestern Amazonia (this study) | 12 | 1.2 | |
| Peatland | High-latitude northern peatlands[29] | 156 | 14 | NMR |
| Forest (non-*Terra preta*) | Amazon Basin (0–30 cm)[26] | 1.4 | 4.7 | HyPy |
| | Amazon Basin (0–100 cm)[26] | 3.6 | 7.3 | |
| Forest (ombrophilous, deciduous and semideciduous) | Northern Brazilian Amazon (0–100 cm)[40] | 1.1 | 1.9 | CTO-375 |
| Forest | Southern Amazonia (0–30 cm)[28] | 2.3 | 4.3 | HyPy |
| Forest fragments | Forest-savanna matrix in Roraima, Brazil (0–100 cm)[27] | 1.4 | 1.3 | CTO-375 |

*HyPy hydrogen pyrolysis, NMR $^{13}$C nuclear magnetic resonance spectroscopy, CTO-375 chemo-thermal oxidation method.

ages covered. The final model presents no significant violations of the analytical assumptions, as shown in the residual errors and Q–Q plots (Supplementary Fig. 5).

### Timing of peat initiation across the Amazon Basin

To assess the timing of peatland initiation across the Amazon Basin, a total of 70 radiocarbon dates from 52 sites were gathered from published sources and our study sites (Supplementary Table 2). A 1 km buffer for each site was generated using the Buffer tool in ArcGIS Pro 3.1.0 (ESRI Inc.) to reduce the influence of sites referring to the same peatland (within the buffer zone). A single average age was used for sites located in the same peatland area. For Ogle Bridge in Guyana, where the calibrated age was reported[79], we used it as provided in that publication. For Laguna Huatacocha in Peru, we calibrated the radiocarbon date using the SHCal20 curve due to its location. All other basal dates were calibrated using the same methods as our radiocarbon dates with a mixed calibration curve in CALIB 8.2[70], and with the SHZ3 post bomb curve in CALIBomb[80]. All calibration ages were reported as median probability.

### Estimation of the PyC stock and associated uncertainty

Since measured PyC was reported as PyC/TOC, a total stock of organic carbon is required for the entire Amazonian peatlands to convert this ratio to a PyC stock. We compiled related peat properties (peat depth, dry bulk density and carbon content, Supplementary Table 1) from publications to calculate the TOC stock for Amazonian peatlands using Eq. (1)[7]. To include sufficiently robust data, we retained only sites with peat thickness >30 cm, and used sites defined as peatlands by authors or that could be shown to be peat from the available geochemical data, as LOI > 30% or C% > 15%, which is the same criterion that applied to our sites.

$$\text{TOC}_\text{p}(\text{Pg}) = A_\text{P} \times D_\text{P} \times \text{DBD}_\text{P} \times C_\text{P}/1000 \quad (1)$$

Where $\text{TOC}_\text{p}$ is the total organic carbon pool (Pg, Pg = $10^{15}$ g), $A_\text{P}$ is peatland area (km$^2$), $D_\text{P}$ is peat depth (m), $\text{DBD}_\text{P}$ is dry bulk density (g cm$^{-3}$), $C_\text{P}$ is carbon content and expressed as a fraction. Peat depth, dry bulk density and carbon content were obtained from the literature, and the peatland area was calculated based on PEATMAP[39].

Then, the PyC stock was estimated using Eq. (2) by calculating PyC stock for each age class separately, as they have different PyC/TOC, and then summing them. Given the lack of comprehensive regional compilations of TOC in the Amazon Basin and limited information as to its vertical distribution, we have assumed that TOC is equally distributed across the age classes. Additionally, instead of using a weighted peat age (age with relative frequency) or a simple average age from our collected peat basal ages, we chose to use the median age to represent a basin-wide peat initiation time to

reduce the bias from very old dates (for examples, Mera and Ogle Bridge), and then infer cumulative PyC stock along the peat profile using this median age.

$$\text{PyC}_\text{p}(\text{Pg}) = \sum_{i=0}^{n}\left(\text{TOC}_P \times \frac{\text{PyC/TOC}}{100}\right) \quad (2)$$

Where $\text{PyC}_\text{p}$ is the total pyrogenic carbon pool (Pg), $\text{TOC}_\text{p}$ is the total organic carbon pool, PyC/TOC is the percent of PyC in TOC (%), which changes with peat age, n is the total number of age classes.

The uncertainty of the PyC estimates was calculated using the law of error propagation in Eq. (3). In this process, we consider the uncertainties from peat depth, carbon content and PyC/TOC. The uncertainty from bulk density is not included because this parameter is closely associated with carbon content[81,82]. The uncertainty of peat area is also not included, as is the case in the calculations of TOC stock.

$$\sigma_{\text{PyC}_\text{p}} = \text{PyC}_\text{p} \times \sqrt{\left(\frac{\sigma_{D_\text{p}}}{D_\text{P}}\right)^2 + \left(\frac{\sigma_{C_\text{p}}}{C_\text{P}}\right)^2 + \left(\frac{\sigma_{\text{PyC/TOC}}}{\text{PyC/TOC}}\right)^2} \quad (3)$$

Where $\text{PyC}_\text{p}$ is the total pyrogenic carbon pool (Pg), $\sigma_{\text{PyC}_\text{p}}$ is its related standard deviation, $\sigma_{D_\text{p}}$ and $\sigma_{C_\text{p}}$ are standard deviations for $D_\text{P},$ and $C_\text{P},$ respectively. $\sigma_{\text{PyC/TOC}}$ is the uncertainty obtained from the maximum values of upper and minimum values of lower confidence levels of the PyC-age model across all age classes (*bootMer* function in the *lme4* package[76]).

### Statistical analysis

Kruskal–Wallis rank sum test (non-parametric test) was used to determine if there were statistically significant differences between sites for both TOC and PyC/TOC, as neither is normally distributed. A non-parametric Dunn's post hoc test was applied to check the influence of site location and type on PyC/TOC. All analysis was conducted using R 4.3.0[75]. Spatial autocorrelation analysis (Moran's I) was used to assess spatial patterns of mean TOC, mean PyC/TOC, and the relationship (slope from simple linear regression model) between PyC/TOC and age among our sites based on Euclidean distance, using South America Albers Equal Area Conic as the projected coordinate system in ArcGIS Pro 3.1.0 (ESRI Inc.). While Global Moran's I Index was calculated using the tool Spatial Autocorrelation (Global Moran's I), Local Moran's I Index was then calculated to identify spatial clusters using the tool Cluster and Outlier Analysis (Anselin Local Moran's I) when there is a significant relation in Global Moran's I Index.

### Reporting summary

Further information on research design is available in the Nature Portfolio Reporting Summary linked to this article.

## Data availability

Data supporting the findings of this study are present in the paper and the Supplementary Materials. Datasets to generate the main figures and long tables are publicly available at https://doi.org/10.6084/m9.figshare.29643533.

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

## Acknowledgements

This work was supported by a PhD scholarship via the QUEX Institute (The University of Queensland and The University of Exeter) for Yuwan Wang. The $^{14}$C dating samples (four samples with code SUERC-103538 to SUERC-103541) were obtained with the support of the National Environmental Isotope Facility (NEIF) under grant NE/S011587/1 (allocation number 2353.0321). Angela V. Gallego-Sala acknowledges funding from the European Research Council (ERC) under the European Union's Horizon 2020 research and innovation programme (grant agreement no. 865403). Juan C. Benavides acknowledges the Partnerships for Enhanced Engagement in

Research (PEER) grant (FAIN AID-OAA-A-11-00012). Eurídice N. Honorio Coronado acknowledges funding from her NERC Knowledge Exchange Fellowship (grant NE/V018760/2). Ted R. Feldpausch acknowledges funding from the UK Natural Environment Research Council (NERC grants NE/W001691/1, NE/R017980/1, NE/N011570/1). Permission to conduct research in Peru is granted through permits from the National Service of Natural Areas Protected by the State (SERNANP) and the National Forestry and Wildlife Service (SERFOR) for the MONANPERU project. We thank Tim Baker, leader of the MONANPERU project, Miguel Pedraza and Carlos Villacorta for their contributions to the fieldwork and Lidiany Carvalho for her assistance in organizing the fieldwork. For the purpose of open access, the author has applied a 'Creative Commons Attribution (CC BY) licence to any Author Accepted Manuscript version arising from this submission'.

## Author contributions

T.R.F., A.G.-S., and Y.W. designed the study. J.C.B., E.N.H.C., Y.W., T.R.F., and A.G.-S. obtained peat samples. Y.W. conducted hydrogen pyrolysis analysis with assistance from M.I.B. Y.W. analysed data and interpreted the results with input from T.R.F. and A.G.-S. Y.W. wrote the paper, with revisions and feedback from A.G.-S., T.R.F., M.I.B., P.M., H.A.M., J.C.B., and E.N.H.C. at all stages.

## Competing interests

The authors declare no competing interests.
