## [Transparent Peer Review file · Communications Earth & Environment]

Wildfire legacies on pyrogenic carbon stocks in Amazonian peatlands

Corresponding Author: Ms Yuwan Wang

Version 0:

Decision Letter:

Dear Ms Wang,

Your manuscript titled "Wildfire legacies on pyrogenic carbon stocks in Amazonian peatlands" has now been seen by 2 reviewers, and we include their comments at the end of this message. They find your work of interest, but some important points are raised. We are interested in the possibility of publishing your study in Communications Earth & Environment, but would like to consider your responses to these concerns and assess a revised manuscript before we make a final decision on publication.

We therefore invite you to revise and resubmit your manuscript, along with a point-by-point response that takes into account the points raised. Please highlight all changes in the manuscript text file.

Please submit your point-by-point responses as a separate file, distinct from your cover letter where you can add responses to the Editors' comments that you do not want to be made available to the reviewers. Word files are preferred. We recommend that any figures, tables or graphs that are included in the response to reviewers are also included in the main article or Supplementary Information.

Please use the following link to submit your revised manuscript, point-by-point response to the referees' comments (which should be in a separate document to any cover letter), a tracked-changes version of the manuscript (as a PDF file) and the completed checklist:

Link Redacted

We hope to receive your revised paper within six weeks; please let us know if you aren't able to submit it within this time so that we can discuss how best to proceed. If we don't hear from you, and the revision process takes significantly longer, we may close your file. In this event, we will still be happy to reconsider your paper at a later date, as long as nothing similar has been accepted for publication at Communications Earth & Environment or published elsewhere in the meantime.

Please do not hesitate to contact us if you have any questions or would like to discuss these revisions further. We look forward to seeing the revised manuscript and thank you for the opportunity to review your work.

Best regards,

Somaparna Ghosh, PhD
Associate Editor,
Communications Earth & Environment
Consulting Editor,
Communications Sustainability

EDITORIAL POLICIES AND FORMATTING

Editorial Policy: [Policy requirements](https://www.nature.com/documents/nr-editorial-policy-checklist.pdf) (Download the link to your computer as a PDF.)

- Behavioural and social science
- Ecological, evolutionary & environmental sciences
- Life sciences

<https://www.nature.com/documents/nr-reporting-summary.zip>

Furthermore, please align your manuscript with our format requirements, which are summarized on the following checklist: [Communications Earth & Environment formatting checklist](https://www.nature.com/documents/commsj-phys-style-formatting-checklist-article.pdf)

and also in our style and formatting guide [Communications Earth & Environment formatting guide](https://www.nature.com/documents/commsj-phys-style-formatting-guide-accept.pdf) .

***** DATA:** Communications Earth & Environment endorses the principles of the Enabling FAIR data project (<http://www.copdess.org/enabling-fair-data-project/>). We ask authors to make the data that support their conclusions available in permanent, publically accessible data repositories. (Please contact the editor if you are unable to make your data available).

All Communications Earth & Environment manuscripts must include a section titled "Data Availability" at the end of the Methods section or main text (if no Methods). More information on this policy, is available at <http://www.nature.com/authors/policies/data/data-availability-statements-data-citations.pdf>.

If a community resource is unavailable, data can be submitted to generalist repositories such as [figshare](https://figshare.com/) or [Dryad Digital Repository](http://datadryad.org/). Please provide a unique identifier for the data (for example a DOI or a permanent URL) in the data availability statement, if possible. If the repository does not provide identifiers, we encourage authors to supply the search terms that will return the data. For data that have been obtained from publically available sources, please provide a URL and the specific data product name in the data availability statement. Data with a DOI should be further cited in the methods reference section.

<http://www.nature.com/authors/policies/availability.html>.

REVIEWER COMMENTS:

Reviewer #1 (Remarks to the Author):

This study investigates the contribution of pyrogenic carbon (PyC) to total organic carbon (TOC) stocks in Amazonian peatlands. The authors analyzed several peat sediment cores for TOC and PyC, investigated PyC variability over time and then combined their dataset with published sources to estimate pyrogenic carbon stocks in the Amazonian peatlands.

This study is relevant as it contributes to the understanding of global carbon stocks and how to manage them. It is novel as it utilizes an arising geochemical technique to quantify PyC and applies it to peatland ecosystems, which are currently

receiving more attention due to their high potential to store soil carbon. In addition, the study also shows dynamics in peat fire regimes and which factors are having an impact. As I am currently doing a similar study in southwestern Australia (estimating TOC stocks in SWA peatlands from a selection of sites/cores), I found this research paper to be a helpful guide on how to approach a study like this. The authors estimate TOC and PyC stocks in a simplified but reasonable way, and there is also sufficient detail on relatively high uncertainties related to such estimates.

One thing I did not understand was the focus on peat age rather than peat depth throughout the study. At the beginning of the results section, I found myself wondering why the authors jump straight to PyC in relationship to age. When estimating carbon stocks, it makes sense to me to look rather at depth than time. The peat age then helps with interpretation of under what conditions these stocks formed and what implications arise from that (for example how much carbon peatlands sequester over time or if there were times of more PyC accumulation due to more fires in the landscape). The authors do point out that that TOC values were relatively stable across the vertical profile. This could be shown in more detail. It was also not clear to me what the total depths of the cores were and how many samples were analyzed for PyC altogether.

However, this is a good and publishable manuscript, easy to follow and relevant to the field. I would just suggest that if it is within the scope of this study, to clarify where peat age is relevant, and to add some detail on the relationship between TOC/PyC and depth. Some more details on the peat sediment itself could also improve readability of this manuscript.

Line 43 necessarily instead necessity?

Line 244 Is it peat age or peat inception age?

Fabian Boesl

Reviewer #2 (Remarks to the Author):

The manuscript by Wang et al. is well written and data is sound as is statistics, So eventually I would like to see: "Wildfires legacies on pyrogenic carbon stocks in Amazonian peatlands" being published in Communications Earth & Environment. However, in its current state I find to be not as attractive as it could be to encourage other scientist to measure pyrogenic carbon (PyC). I myself was thinking of measuring it in the future, but now I think it is included when measuring total organic carbon (TOC). TOC is a crude bulk C measurement consisting of many differently persistent soil carbon fractions not just PyC. So why should I care for PyC as I still have the amounts of soil carbon measured right? Yes, because we can draw conclusions about processes of production and decay of PyC and TOC. It will improve our modelling of future scenarios of the fate of peatland carbon, but the authors are not stressing that enough. They rather stick to describing dutifully what they observed. I highly encourage them to argue more in a process related manner. For example, are the patterns they found (lower and higher amounts of top soil PyC) related to fire intensities or number of fires in the closer surrounding (satellites images can help here) or more to the vegetation. This can be also seen for boreal/subpolar peatlands where (at least to the pictures I have seen) fires are way more abundant than in the Amazon (which makes sense): I know the authors referred to this a bit but not enough. It also quite clear that PyC relatively increases to TOC with depth, but do the difference found between the sites tell us anything? At some sites PyC could be less persistent than elsewhere or is it really the production side? This applies to literature data compared to own as well as only for the data within the northwestern Amazon. Therefore I recommend the editor major revision.

Some detailed examples:

Abstract Lines 29 – 33: "Estimates of PyC stock in peatland ecosystems exceed those in Amazonian forest soils but are lower than those in high-latitude northern peatlands. Our findings indicate the importance of PyC generated by past fires for peatland ecosystems and highlight its potential long-term role in the future carbon cycle."

It was to be expected (higher than in forests but lower than in northern peatlands) so why does it indicate the importance and its potential long term role?

Line 110-111: "our current understanding of Amazonian peatland ecosystems in the region is in its infancy."

This is too bold as we know a lot of these ecosystems, but maybe not enough on its carbon turnover?

My critique mainly relates to the question the authors raised in lines 120 – 122: "What is the vertical distribution of PyC in Amazonian peatlands? What is the PyC stock in Amazon peatlands? Does PyC storage in peat soils differ from forest soils in the Amazon Basin, considering the area of each soil type??"

The question what is the PyC stock is justified and should be the first question but not what is the distribution. In my opinion it should be ask do we find differences in vertical distributions and if so what do they tell us. I would also raise the question about the horizontal (top soil or layers of the same age) distribution across the Amazon – can we draw any conclusions out of that. Last but not least of course is PyC storage in peatland higher than in forests but do we see a spatial pattern relating them so do high concentrations in forest occur at similar spots as high concentrations in peatlands? Are higher concentration related to higher fire frequencies or intensities - if not the patterns may show differences in PyC decay rates?

Lines 146- 147 suggesting that PyC/TOC is effectively preserved in older, deeper peat layers.

Fair enough but but are there spatial differences that may tell us something?

Lines 189 - 191 "Our analysis of PyC storage in Amazonian peatlands reveals a much higher PyC stock than in forest soils in the basin, even in northwestern Amazonia, where fires have been rare"

Why starting the discussion like that but that is not the main point. So if fires are low in the northwestern Amazon they are low in forest too? So again use your spatial disaggregation into different zones of Amazonia and see if you find different patterns.

Lines 188 – 202 Our findings highlight the significant sequestration of PyC in carbon-rich peatland ecosystems 200 and

suggest the necessity of incorporating PyC in peatlands into regional and global carbon budgets. True but why? They are included in budgets based on TOC Measurements but for scenarios modelling we should be able to differentiate into differently persistent soil carbon and PyC is one prominent example - more importantly the different decay rates of different region of Amazonia need to be considered too (if my assumptions prove right)

Communications Earth & Environment is committed to improving transparency in authorship. As part of our efforts in this direction, we are now requesting that all authors identified as 'corresponding author' create and link their Open Researcher and Contributor Identifier (ORCID) with their account on the Manuscript Tracking System prior to acceptance. ORCID helps the scientific community achieve unambiguous attribution of all scholarly contributions. You can create and link your ORCID from the home page of the Manuscript Tracking System by clicking on 'Modify my Springer Nature account' and following the instructions in the link below. Please also inform all co-authors that they can add their ORCIDs to their accounts and that they must do so prior to acceptance.

Version 1:

Decision Letter:

Dear Ms Wang,

Your manuscript titled "Wildfire legacies on pyrogenic carbon stocks in Amazonian peatlands" has now been seen by our reviewers, whose comments appear below. In light of their advice we are delighted to say that we are happy, in principle, to publish a suitably revised version in Communications Earth & Environment.

We therefore invite you to revise your paper one last time to address the remaining concerns of our reviewers. At the same time we ask that you edit your manuscript to comply with our format requirements and to maximise the accessibility and therefore the impact of your work.

EDITORIAL REQUESTS:

*****Please take care to match our formatting and policy requirements. We will check revised manuscript and return manuscripts that do not comply. Such requests will lead to delays. *****

SUBMISSION INFORMATION:

OPEN ACCESS:

Communications Earth & Environment is a fully open access journal. Articles are made freely accessible on publication. For further information about article processing charges, open access funding, and advice and support from Nature Research, please visit <https://www.nature.com/commsenv/open-access>

Link Redacted

Best regards,

Somaparna Ghosh, PhD
Associate Editor,
Communications Earth & Environment
Consulting Editor,
Communications Sustainability

REVIEWERS' COMMENTS:

Reviewer #1 (Remarks to the Author):

I am satisfied with the revision and the authors' response.

Reviewer #2 (Remarks to the Author):

No further objections - well done - I like using Moron's I.

The responses to the reviewers' comments are in blue, while the revisions made to the manuscript are marked in red in tracked change version. Text cited from the revised manuscript and supplementary information is shown in *italic brown*. Line numbers in this letter refer to the tracked changes version of the revised manuscript.

Reviewer #1 (Remarks to the Author):

This study investigates the contribution of pyrogenic carbon (PyC) to total organic carbon (TOC) stocks in Amazonian peatlands. The authors analyzed several peat sediment cores for TOC and PyC, investigated PyC variability over time and then combined their dataset with published sources to estimate pyrogenic carbon stocks in the Amazonian peatlands.

This study is relevant as it contributes to the understanding of global carbon stocks and how to manage them. It is novel as it utilizes an arising geochemical technique to quantify PyC and applies it to peatland ecosystems, which are currently receiving more attention due to their high potential to store soil carbon. In addition, the study also shows dynamics in peat fire regimes and which factors are having an impact. As I am currently doing a similar study in southwestern Australia (estimating TOC stocks in SWA peatlands from a selection of sites/cores), I found this research paper to be a helpful guide on how to approach a study like this. The authors estimate TOC and PyC stocks in a simplified but reasonable way, and there is also sufficient detail on relatively high uncertainties related to such estimates.

One thing I did not understand was the focus on peat age rather than peat depth throughout the study. At the beginning of the results section, I found myself wondering why the authors jump straight to PyC in relationship to age. When estimating carbon stocks, it makes sense to me to look rather at depth than time. The peat age then helps with interpretation of under what conditions these stocks formed and what implications arise from that (for example how much carbon peatlands sequester over time or if there were times of more PyC accumulation due to more fires in the landscape).

Response #1: Thanks for the comment. In this study, we used peat age as the primary axis for PyC stock estimation, as it provides a consistent framework to explore how PyC accumulation has varied over time, given that peat accumulation rate can vary across time. Peat accumulation rates differ markedly in our sites (max: 0.30 ± 0.08 cm yr⁻¹ in CAN; min: <0.01 cm yr⁻¹ in INI). Additionally, this PyC-age approach makes it easily to facilitate the direct application of our equations and comparison of our findings

to other paleoenvironmental studies, such as in northern peatlands (Leifeld et al., 2018) and link to climatic settings that could affect PyC formation, over similar timescales.

We agree that integrating both depth- and age-based methods could help provide a more comprehensive understanding. Therefore, we followed the suggestion and calculated the PyC stock based on the PyC-depth relationship using the same approach as PyC-age procedure. The result is presented in **Figure S6**. This depth-derived PyC stock, 0.32 ± 0.37 Pg, is more likely to be underestimated compared to the one derived from age (0.73 ± 0.61 Pg), representing only 0.7% of TOC stock for the Amazonian peatlands. This is also much lower compared to the observations in this study (1.2% on average). Therefore, we used PyC-age in this study.

We have added sentences in Supplementary Text and Figure S6 to the supplementary information:

Supplementary Text: “An additional concern in this study is that the estimation of PyC stock is based on its relationship with age instead of depth. This approach aligns with a previous PyC stock estimation in northern peatlands, which also used age as the predictor (Leifeld et al., 2018). Using age also facilitates the direct application of equations and allows linking to climatic settings that could affect PyC formation, over similar timescales. Additionally, PyC estimation based on the PyC-depth relationship yields 0.32 ± 0.37 Pg of PyC for basin-wide peatlands (Figure S6), representing only 0.7% of TOC stock in Amazonian peatlands. This is much lower than the 1.2% average in our observations. Therefore, the age-based estimation is preferred in this study.”

Figure S6 The relationship between the percentage of pyrogenic carbon in total organic carbon and peat depth in Amazonian peatlands (N=67). The shape of data points indicates different peat cores. The solid line is the ‘best’ model derived from a linear mixed-effects model with a random slope (equation insert). Dashed lines indicate

a 95% confidence interval. This gives the estimation of 0.32 ± 0.37 Pg PyC for basin-wide peatland.

The authors do point out that that TOC values were relatively stable across the vertical profile. This could be shown in more detail. It was also not clear to me what the total depths of the cores were and how many samples were analyzed for PyC altogether.

Response #2: Thanks for the comment. We have now added the relationship between TOC and peat age/depth in **Figure S7** and the detailed description of the core and samples is added in **Table S3**.

We have added Figure S7 and Table S3 to the supplementary information:

Figure S7 The relationship between TOC and (a) depth and (b) age in study sites. The shape of data points indicates different peat cores.

Table S3 Detailed Information of study sites and the number of PyC samples.

Site	Lat (°)	Lon (°)	Core length (cm)	Core description	Core length included in the study (cm)	No. of PyC samples in peat section
Inírida (INI)	3.77	-67.93	100	Homogenous peat throughout the core.	42.5	5
Puerto Lleras (PLL)	3.26	-73.42	500	Homogeneous peat until 220 cm with clay intrusion between 20 and 38 cm; brown-grey sediment with peat stripes to 280 cm; dark peat with	300	11

				several clay/silt stripes from 280 to 500 cm.		
Caquetá (CAQ)	1.13	-75.43	400	Peat until 50 cm; grey sediment between 50 and 120 cm; dark-brown peat with several thin grey stripes between 120 and 315 cm; grey sediment from 315 to 400 cm.	400	12
Quistococha (QUI)	-3.83	-73.32	300	Homogenous peat throughout the core.	300	16
Jenaro Herrera (JEN)	-4.96	-73.67	24	Homogenous peat throughout the core.	24	5
Cananguchal (CAN)	-3.81	-70.23	300	Homogeneous peat until 190 cm; gradual transition from peat to clay-rich sediment between 190 and 196 cm; yellowish clay materials from 196 to 300 cm.	200	18

However, this is a good and publishable manuscript, easy to follow and relevant to the field. I would just suggest that if it is within the scope of this study, to clarify where peat age is relevant, and to add some detail on the relationship between TOC/PyC and depth. Some more details on the peat sediment itself could also improve readability of this manuscript.

Response #3: Thanks for the suggestion. We have clarified the use of peat age in **Line 114-120**. The estimation of PyC stock based on PyC-depth is shown in **Figure S6** (see **Response #1**). The relationship between TOC and depth/age and the description of peat core is addressed in **Response #2**.

Line 114-120: *“The only study of PyC stored in peatlands is from high-latitude northern peatlands and reported a total PyC stock of ~62 Pg based on PyC-age relationships²⁹. The age of the peat provides information on long-term apparent rates of carbon*

accumulation. This facilitates the temporal comparison of carbon dynamics for large datasets on the regional scale ³⁰ and possibly allows linking broader climatic controls at the same temporal scale ³¹.”

Line 43 necessarily instead necessity?

Response #4: Changed as suggested (**Line 51**).

Line 244 Is it peat age or peat inception age?

Response #5: Sorry for the confusion, it should be peat age here, the word has been changed accordingly (**Line 305**).

Fabian Boesl

Reviewer #2 (Remarks to the Author):

The manuscript by Wang et al. is well written and data is sound as is statistics, So eventually I would like to see:” Wildfires legacies on pyrogenic carbon stocks in Amazonian peatlands” being published in Communications Earth & Environment. However, in its current state I find to be not as attractive as it could be to encourage other scientist to measure pyrogenic carbon (PyC). I myself was thinking of measuring it in the future, but now I think it is included when measuring total organic carbon (TOC). TOC is a crude bulk C measurement consisting of many differently persistent soil carbon fractions not just PyC. So why should I care for PyC as I still have the amounts of soil carbon measured right? Yes, because we can draw conclusions about processes of production and decay of PyC and TOC. It will improve our modelling of future scenarios of the fate of peatland carbon, but the authors are not stressing that enough. They rather stick to describing dutifully what they observed. I highly encourage them to argue more in a process related manner. For example, are the patterns they found (lower and higher amounts of top soil PyC) related to fire intensities or number of fires in the closer surrounding (satellites images can help here) or more to the vegetation. This can be also seen for boreal/subpolar peatlands where (at least to the pictures I ave seen) fires are way more abundant than in the Amazon (which makes sense): I know the authors refered to this a bit but not enough. It also quite clear that PyC relatively increases to TOC with depth, but do the difference found between the sites tell us anything? At some sites PyC could be less persistent than elsewhere or is it really the production side? This applies to literature data compared to own as well as only for the data within the northwestern Amazon. Therefore I recommend the editor major revision.

Response #6: We thank Reviewer 2 for the constructive comments, particularly for the valuable suggestions regarding the spatial analysis, which have strengthened the overall quality of the manuscript. We have summarised the major changes made in the revised manuscript, with further point-by-point responses provided below.

- The research questions have been revised for improved clarity.
- The role of PyC in carbon dynamics models has been more clearly highlighted. The spatial variability of PyC at the study sites has been addressed by adding a new section titled “**Spatial variability of pyrogenic carbon**” in the **Discussion (Line 486-535)**.

Some detailed examples:

Abstract Lines 29 – 33: “Estimates of PyC stock in peatland ecosystems exceed those

in Amazonian forest soils but are lower than those in high-latitude northern peatlands. Our findings indicate the importance of PyC generated by past fires for peatland ecosystems and highlight its potential long-term role in the future carbon cycle.”

It was to be expected (higher than in forests but lower than in northern peatlands) so why does it indicate the importance and its potential long term role?

Response #7: Thanks for the comment. We have revised the sentences to better clarify the role of PyC due to its long residue time (**Line 26-29**).

Line 26-29 *“Due to the slower turnover of PyC in peatland ecosystems, our findings indicate the importance of PyC generated by past fires for these ecosystems in the global PyC budget. This highlights potential long-term carbon sequestration role of PyC in the future carbon cycle.”*

Line 110-111: “our current understanding of Amazonian peatland ecosystems in the region is in its infancy.”

This is too bold as we know a lot of these ecosystems, but maybe not enough on its carbon turnover?

Response #8: Thanks for the comments. We have changed the word in **Line 127-131**.

Line 127-131: *“Although significant progress has been made in understanding carbon cycling in lowland Amazonian peatlands, such as those in Peru based on intensive fieldwork ³⁶, our understanding of specific processes, such as PyC accumulation and dynamics is very limited.”*

My critique mainly relates to the question the authors raised in lines 120 – 122: “What is the vertical distribution of PyC in Amazonian peatlands? What is the PyC stock in Amazon peatlands? Does PyC storage in peat soils differ from forest soils in the Amazon Basin, considering the area of each soil type??

The question what is the PyC stock is justified and should be the first question but not what is the distribution. In my opinion it should be ask do we find differences in vertical distributions and if so what do they tell us.

Response #9: Thanks for the comment. The reason why the questions are listed in this order is that the first question is for the studied peatlands and the second is for the basin-wide peatlands based on the relationship from the first question. We have modified the questions for better flow and understanding as follows:

Line 144-149: *“This study addresses the following questions: (a) Is there an age-*

dependent vertical distribution of PyC in the studied peatlands, and if so, what is the pattern? (b) What is the PyC stock in basin-wide peatlands, representing the total PyC accumulated during peat formation in the Amazon Basin? (c) Does PyC storage in peat soils differ from forest soils in the Amazon Basin, considering the area of each soil type?”

I would also raise the question about the horizontal (top soil or layers of the same age) distribution across the Amazon – can we draw any conclusions out of that.

Response #10: Thanks for the comment. To assess the spatial patterns in the study sites, Moran's I index has now been introduced in **Line 739-748**. We used the mean PyC values in a single peat profile to represent the general fire history of each site. Moran's I index is a widely used method to detect spatial autocorrelation based on geographical locations (Petroni et al., 2004), normally ranging from -1 (negative spatial autocorrelation, dispersed) to 1 (positive spatial autocorrelation, clustered), while a local index of Moran's focuses on each site, identifying where are the clusters and outliers (Anselin, 1995) in PyC/TOC distribution.

Line 739-748: *“Spatial autocorrelation analysis (Moran's I) was used to assess spatial patterns of mean TOC, mean PyC/TOC, and the relationship (slope from simple linear regression model) between PyC/TOC and age among our sites based on Euclidean distance, using South America Albers Equal Area Conic as the projected coordinate system in ArcGIS Pro 3.1.0 (ESRI Inc.). While Global Moran's I Index was calculated using the tool Spatial Autocorrelation (Global Moran's I), Local Moran's I Index was then calculated to identify spatial clusters using the tool Cluster and Outlier Analysis (Anselin Local Moran's I) when there is a significant relation in Global Moran's I Index.”*

The result of this analysis is added in **Line 162-165** and in **Figure S1-2**. There are also significantly lower PyC values in the south sites compared to the northern sites in all observations (**Figure S3**), which also proves the existence of a spatial difference. Further detailed discussion on spatial variability is included in **Spatial variability of pyrogenic carbon** section (**Line 486-535**).

Line 162-165: *“Significant spatial autocorrelation was found in PyC/TOC (Figure S1a), while a cluster of low values appeared to be located at southern sites (QUI, JEN and CAN) (Figure S2). No spatial autocorrelation in TOC was found (Figure S1b).”*

We have added Figure S1-3 to the supplementary information.

Figure S1 Global Moran's index and related p-values of (a) PyC/TOC, (b) TOC and (c) the relationship between PyC/TOC and age (slope from simple linear regression model) based on different thresholds of Euclidean distance in Global Moran's I test. Dashed line marks 0.05 of p-values.

Figure S2 Local Moran's *I* maps of PyC/TOC (%) at Euclidean distance thresholds where Global Moran's *I* was significantly positive, based on the Global Fire Emissions Database with small fires (GFEDs) burned fraction over the period of 1997-2016 (Van Der Werf et al., 2017).

Figure S3 Percentage of pyrogenic carbon in total organic carbon by (a) site location (North: INI, PLL and CAQ; South: QUI, JEN and CAN) and (b) site type (flooded-flooded forest: INI; mixed P-mixed palm swamp: CAQ; P(agri): palm swamp surrounded by agricultural land: PLL and denseP-Mauritia-dominated palm swamp: QUI, JEN and CAN). $p < 0.01$ from Kruskal-Wallis test for site location and type. The different letters on the top in (b) indicate significant differences (Dunn's post hoc test).

Last but not least of course is PyC storage in peatland higher than in forests but do we see a spatial pattern relating them so do high concentrations in forest occur at similar spots as high concentrations in peatlands? Are higher concentration related to higher fire frequencies or intensities - if not the patterns may show differences in PyC decay rates?

Response #11: Thanks for the comment. Due to the lack of direct geographical comparison of data between peatlands and forest soils, and given no spatial autocorrelation was detected in PyC/TOC in forest soils (0-5 cm) in a basin-wide study in forest soils (Koele et al., 2017), we are currently unable to draw a certain conclusion on this. Considering that PyC is the byproduct of wildfires, it is reasonable that we would expect high PyC values in forests to tend to co-occur spatially with nearby peatlands, as they would share the similar climatic settings on a regional scale; for example, the lower PyC values are more likely to be present in the wet climate of northwestern Amazonia compared to the other regions in the basin. If such spatial patterns exist, they could indicate the broader climatic control on regional fire regimes and this is based on the assumptions of limited human activities (McMichael et al., 2012). However, for the forest areas where intensive human activities or Anthropogenic dark earth soils are present, the intentionally enriched soils could contain elevated PyC levels (WinklerPrins et al., 2020).

In this study, we used hydrogen pyrolysis to quantify PyC, and this technique could isolate the stable fraction of PyC with ring size greater than seven (Meredith et al., 2012) and this pool is assumed to have a half-life from centuries to millennia (Bird et al., 2015). PyC_{hypy} is generally produced in greater amounts during fires with higher temperatures, which favour the formation of more condensed structures (Ascough et al., 2020; Bird et al., 2024); therefore, PyC in this study reflects more intense burning events, as shown in this study (**Figure S2**) with clusters of lower PyC values in southern sites. This suggests that the production of PyC influences the PyC values and related discussion on the fire regimes and ecosystem types of sites are as follows:

Line 511-535: *“Yearly fire burn fraction (per grid cell) between 1997 and 2016, from Global Fire Emissions Database with small fires (GFEDs)⁶⁶ shows more frequent burning in our northern sites and surrounding areas that are closer to the edge of the basin than QUI, JEN and CAN, where the lower clusters of PyC/TOC were observed (Figure S2). This suggests peatlands that are surrounded by drier environments at the edge of the Amazon Basin (e.g., areas adjacent to the Cerrado or Llanos) potentially receive higher PyC inputs from nearby fire-prone environments, which contribute to higher PyC deep storage at those sites.*

Mauritia-dominated palm swamps of three southern sites (QUI, JEN and CAN), had

presented significantly lower PyC values compared to all the other ecosystem types in this study, including mixed palm swamps, palm swamps and flooded forest (Figure S3). It is worth noting that PyC/TOC values in the INI site are much higher than in the other sites, which may partly be explained by this site experiencing more variable water levels in flooded forest than in palm swamps⁶⁷. Therefore, there is greater potential for being affected by lower water levels during droughts or prolonged dry periods, which would increase the likelihood of local fires occurring.”

However, without direct decay rate measurements in the study sites, we cannot rule out that some variability arises from site-specific degradation. Based on our current sites, there is no spatial correlation found in the vertical distribution of PyC values (**Line 181-182**), future work pairing PyC with decay trials would refine these interpretations. We have added sentences to address this for future research in **Line 478-484**.

Line 181-182: *No spatial autocorrelation in the trend of PyC/TOC was observed (Figure S1c).”*

Line 478-484: *“The lack of a spatial pattern in PyC/TOC over time in our study sites may indicate that each site is differently influenced by local conditions (e.g., hydrology, topography, remote PyC sources, and peat accumulation rates) and these local factors are more important than regional phenomena in driving PyC spatial patterns. This issue needs to be investigated in future research to gain a clearer understanding of main drivers of PyC accumulation.”*

Lines 146- 147 suggesting that PyC/TOC is effectively preserved in older, deeper peat layers.

Fair enough but but are there spatial differences that may tell us something?

Response #12: Thanks for the comment. We tested the slope of PyC/TOC with age using Moran's I index and there is no significant spatial pattern among the study sites (see **Response #11**). For individual-site results, PyC/TOC in two of all sites showed a significant increase (**Table 1**); the mechanism behind this trend is not well-defined. A clearly spatial pattern and the drivers of PyC/TOC changes through time will require additional sites from the same region or the whole basin, something that could be addressed in future research.

Lines 189 - 191 “Our analysis of PyC storage in Amazonian peatlands reveals a much higher PyC stock than in forest soils in the basin, even in northwestern Amazonia, where fires have been rare”

Why starting the discussion like that but that is not the main point. So if fires are low in the northwestern Amazon they are low in forest too? So again use your spatial disaggregation into different zones of Amazonia and see if you find different patterns.

Response #13: Thanks for the comment. The first paragraph in the Discussion section has been modified in the revised manuscript between **Line 225 and 260**. We revised sentences mentioned as follows in **Line 235-255**.

Line 235-255: *“Our analysis of PyC storage in Amazonian peatlands reveals a much higher PyC stock per unit area than in forest soils in the basin. The calculations of basin-wide PyC peatland stocks are based on northwestern Amazonia, where fires have been rare due to very wet conditions and limited human intervention; thus, this storage estimate may be an underestimation.”*

Lines 188 – 202 Our findings highlight the significant sequestration of PyC in carbon-rich peatland ecosystems 200 and suggest the necessity of incorporating PyC in peatlands into regional and global carbon budgets.

True but why? They are included in budgets based on TOC Measurements but for scenarios modelling we should be able to differentiate into differently persistent soil carbon and PyC is one prominent example - more importantly the different decay rates of different region of Amazonia need to be considered too (if my assumptions prove right)

Response #14: Thanks for the comment. We revised the sentences to better explain the importance of PyC in future research in **Line 255-260**.

Line 255-260: *“Our findings highlight the significant sequestration of PyC in carbon-rich peatland ecosystems and suggest the necessity of incorporating PyC in peatlands into regional and global carbon dynamic models. This is particularly the case because the residence time of PyC in peatlands (and mineral soils) is very different from non-PyC fractions in TOC and this needs to be explicitly taken into account in these models.”*

Reference:

- Anselin, L. (1995). Local indicators of spatial association—LISA. *Geographical analysis*, 27(2), 93-115.
- Ascough, P. L., Brock, F., Collinson, M. E., Painter, J. D., Lane, D. W., & Bird, M. I. (2020). Chemical characteristics of macroscopic pyrogenic carbon following millennial-scale environmental exposure. *Frontiers in Environmental Science*, 7, 203.
- Bird, M. I., Brand, M., Comley, R., Fu, X., Hadeen, X., Jacobs, Z., . . . Bradshaw, C. J. (2024). Late Pleistocene emergence of an anthropogenic fire regime in Australia’s tropical

- savannahs. *Nature Geoscience*, 17(3), 233-240.
- Bird, M. I., Wynn, J. G., Saiz, G., Wurster, C. M., & McBeath, A. (2015). The pyrogenic carbon cycle. *Annual Review of Earth and Planetary Sciences*, 43(1), 273-298.
- Koele, N., Bird, M., Haig, J., Marimon-Junior, B. H., Marimon, B. S., Phillips, O. L., . . . Feldpausch, T. R. (2017). Amazon Basin forest pyrogenic carbon stocks: First estimate of deep storage. *Geoderma*, 306, 237-243.
- Leifeld, J., Alewell, C., Bader, C., Krüger, J. P., Mueller, C. W., Sommer, M., . . . Szidat, S. (2018). Pyrogenic carbon contributes substantially to carbon storage in intact and degraded northern peatlands. *Land Degradation & Development*, 29(7), 2082-2091.
- McMichael, C. H., Piperno, D. R., Bush, M. B., Silman, M. R., Zimmerman, A. R., Raczka, M. F., & Lobato, L. C. (2012). Sparse pre-Columbian human habitation in western Amazonia. *Science*, 336(6087), 1429-1431.
- Meredith, W., Ascough, P., Bird, M. I., Large, D., Snape, C., Sun, Y., & Tilston, E. (2012). Assessment of hydropyrolysis as a method for the quantification of black carbon using standard reference materials. *Geochimica et Cosmochimica Acta*, 97, 131-147.
- Petrone, R. M., Price, J., Carey, S., & Waddington, J. (2004). Statistical characterization of the spatial variability of soil moisture in a cutover peatland. *Hydrological Processes*, 18(1), 41-52.
- Van Der Werf, G. R., Randerson, J. T., Giglio, L., Van Leeuwen, T. T., Chen, Y., Rogers, B. M., . . . Kasibhatla, P. S. (2017). Global fire emissions estimates during 1997-2016. *Earth System Science Data*, 9(2), 697-720.
- WinklerPrins, A. M. A., & Levis, C. (2020). Reframing Pre-European Amazonia through an anthropocene lens. *Annals of the American Association of Geographers*, 1-11.